# Analysis and differentiation of tobacco-derived and synthetic nicotine products: Addressing an urgent regulatory issue

**Andrew G. Cheetham** [1]*, **Susan Plunkett**[1,2], **Preston Campbell**[2], **Jacob Hilldrup**[1], **Bonnie G. Coffa**[1], **Stan Gilliland, III** [2], **Steve Eckard**[1]

**1** Enthalpy Analytical, LLC, Richmond, Virginia, United States of America, **2** Consilium Sciences, LLC, Richmond, Virginia, United States of America

* andrew.cheetham@enthalpy.com

**Data Availability Statement:** All relevant data are within the paper and its Supporting Information files.

## Abstract

There is significant regulatory and economic need to distinguish analytically between tobacco-derived nicotine (TDN) and synthetic nicotine (SyN) in commercial products. Currently, commercial e-liquid and oral pouch products are available that contain tobacco-free nicotine, which could be either extracted from tobacco or synthesized. While tobacco products that contain TDN are regulated by FDA Center for Tobacco Products, those with SyN are currently not in the domain of any regulatory authority. This regulatory difference provides an economic incentive to use or claim the use of SyN to remain on the market without submitting a Premarket Tobacco Product Application. TDN is ~99.3% ($S$)-nicotine, whereas SyN can vary from racemic (50/50 ($R$)/($S$)) to $\geq$ 99% ($S$)-nicotine, i.e., chemically identical to the tobacco-derived compound. Here we report efforts to distinguish between TDN and SyN in various samples by characterizing impurities, ($R$)/($S$)-nicotine enantiomer ratio, ($R$)/($S$)-nornicotine enantiomer ratio, and carbon-14 ($^{14}$C) content. Only $^{14}$C analysis accurately and precisely differentiated TDN (100% $^{14}$C) from SyN (35–38% $^{14}$C) in all samples tested. $^{14}$C quantitation of nicotine samples by accelerator mass spectrometry is a reliable determinate of nicotine source and can be used to identify misbranded product labelled as containing SyN. This is the first report to distinguish natural, bio-based nicotine from synthetic, petroleum-based nicotine across a range of pure nicotine samples and commercial e-liquid products.

## Introduction

### Regulatory context

The Family Smoking Prevention and Tobacco Control Act ("FSPTCA") gives the Food and Drug Administration ("FDA") authority to regulate the manufacture, distribution, and marketing of tobacco products in the United States [1]. The Act defines a tobacco product as "any product made or derived from tobacco that is intended for human consumption, including any component, part, or accessory of a tobacco product (except raw materials other than

**Funding:** The authors received no specific funding for this work.

**Competing interests:** The authors are paid employees of their respective companies. Enthalpy Analytical, LLC, a wholly owned subsidiary of Montrose Environmental Group, Inc., is an independent contract research laboratory with a focus on nicotine-containing products that provides analytical testing services for a wide range of clients, including tobacco manufacturers and government regulatory authorities. Consilium Sciences provides consulting services, offering scientific and regulatory solutions for materials science, nicotine, and cannabis organizations with a focus on potentially harm-reduced products.

tobacco used in manufacturing a component, part, or accessory of a tobacco product).” The FDA Center for Tobacco Products (CTP) regulates any product containing tobacco-derived materials, either leaf tobacco or tobacco-derived nicotine, or nicotine-free products intended to be used as tobacco products. In order to legally market a tobacco product, FDA must issue a marketing granted order for a tobacco product based on a comprehensive, expensive application through the substantial equivalence (SE) or premarket tobacco product application (PMTA) pathways. FDA's Center for Drug Evaluation and Research (CDER) regulates nicotine products intended to be used as a drug, device, or combination product. One example is over-the-counter nicotine replacement therapy (NRT) products in various formats such as gum, patch, lozenge, nasal spray, and tablets. Per CDER's mission, these nicotine drug products must be safe, effective, and therapeutic. There is significant regulatory ambiguity whether synthetic nicotine-containing products are a tobacco product, a drug, or neither. Additionally, whether CTP or CDER has jurisdiction over synthetic nicotine products is unclear despite the identical chemical composition of nicotine sourced from natural tobacco plant-derived versus synthetic processes.

This regulatory difference incentivizes manufacturers attempting to evade regulation to use SyN to remain on the market. Puff Bar is the earliest publicly acknowledged example of a product switching from tobacco-derived nicotine (TDN) to synthetic nicotine (SyN), in response to a July 2020 FDA letter ordering the removal of Puff Bar e-cigarettes from the market for lacking the required premarket authorization [2]. In early 2021, Puff Bar announced they were returning to the market, claiming that their "nicotine-containing products are crafted from a patented manufacturing process, not from tobacco" [3]. The Puff Bar website states that "All Puff Bar products listed on this website contain nicotine but do not contain tobacco or anything derived from tobacco. Puff Bar products are not intended for use with any tobacco product or any component or part of a tobacco product." A study by Duell et al. examined both early and current Puff Bar products, concluding that they did switch from TDN to SyN [4]. The authors found that the older Puff Bar products contained >99% (*S*)-nicotine, whereas the newer SyN-containing Puff Bars contained both (*R*)- and (*S*)-isomers in a ~1:1.2 ratio, inconsistent with a racemic SyN being used (see *Scientific Context* below). Further spotlighting Puff Bar, the 2021 National Youth Tobacco Survey (NYTS) showed that Puff Bar was the most popular brand among youth e-cigarette users [5], stating that, "Among high school current e-cigarette users, 26.1% reported that their usual brand was Puff Bar," and, "Among middle school current users, 30.3% reported that their usual brand was Puff Bar." Beyond FDA CTP attention, the legislative branch is also interested in Puff Bar and SyN. On November 8, 2021, Representative Raja Krishnamoorthi (D-Illinois), the chair of the Subcommittee on Economic and Consumer Policy, requested that Puff Bar provide, among other things, "all documents and communications referring or relating to the use of synthetic nicotine in Puff Bar products, including the decision to switch to synthetic nicotine, and all documents relating to the purchase of synthetic nicotine" [6]. Most recently, Representative Mikie Sherrill (D-New Jersey) introduced the "Clarifying Authority Over Nicotine Act of 2021," bipartisan legislation that would ensure SyN-based products are within FDA CTP's regulatory purview [7].

As of October 13, 2021, FDA CTP had taken action on more than 98% of the over 6.5 million products in electronic nicotine delivery systems (ENDS) PMTAs. These actions were largely marketing denial orders (MDOs) for more than 1 million non-tobacco flavored ENDS products [8]. After receiving MDOs, many companies publicly stated their intent to switch to SyN to remain on the market and avoid business closure [9, 10]. Current FDA CTP Director Mitch Zeller acknowledged that "To try to avoid FDA regulation and evade enforcement, several companies that received MDOs are publicly saying they are switching to synthetic nicotine to keep their products on the market." [11] Mainstream media coverage in outlets such as

Time Magazine [12] and Politico [13] also indicates that this issue has become part of the public consciousness.

It is unclear if companies are actually transitioning to SyN or simply claiming its use to skirt regulations. Companies must carefully consider the financial implications of switching to SyN, as it is currently much more expensive than TDN at roughly four times the cost. Current supply cannot meet the growing demand. Thus, there is economic incentive to claim the use of SyN while actually using the much cheaper TDN or a mixture of the two. Accordingly, there is clear, significant regulatory and economic need to distinguish analytically between TDN, SyN, and their mixtures.

## Scientific context

Nicotine is an optically active molecule, existing as two enantiomers that are denoted as (*S*)-nicotine and (*R*)-nicotine (Fig 1). While the enantiomers have differing pharmacological properties, regulatory authorities do not currently distinguish between the two forms, only between how the nicotine is produced. Fig 2 illustrates the production pathways for both tobacco-derived and synthetic nicotine. Tobacco-derived nicotine is predominately the (*S*)-enantiomer ($\geq$ 99%), with only minor amounts of the (*R*)-enantiomer ($\leq$ 1%). The (*S*)-enantiomer possesses the well-known pharmacological properties associated with tobacco use (Fig 1). SyN is commonly produced as a racemic (50:50) mixture of both the (*S*)- and (*R*)-enantiomers [14, 15] that then may be enriched to produce $\geq$ 99% (*S*)-nicotine [15, 16]. The exception to this is a chemoenzymatic approach patented by Zanoprima Lifesciences [17] that enzymatically reduces myosmine to produce $\geq$ 99% (*S*)-nornicotine, followed by methylation to yield (*S*)-nicotine. More information can be found in a recent article by Jordt [18], who provides a thorough discussion of SyN's history and the various manufacturing pathways. Both the racemic and enriched forms are marketed as SyN, with the racemic mixture being the cheaper option since enantiomeric enrichment is not required. Historically, SyN has been very expensive to purchase due to the aforementioned production needs and little commercial demand relative to the cheaper, more abundant TDN. However, the price of SyN has been decreasing as the demand grows from manufacturers attempting to evade regulation by using SyN to remain on the market. Due to the relatively high cost of SyN and rare use, there has been very little published literature regarding SyN.

Whether extracted from tobacco or synthesized from precursors, the nicotine produced is the same chemical compound. Consequently, the production source cannot be determined by

**(*S*)-Nicotine**
CAS 54-11-5

Well established pharmacology
Agonist at nicotinic acetylcholine receptor (nAChRs)
No direct action on acetylcholinesterase (AChE)
≥ 99% of tobacco-derived nicotine is (*S*)-nicotine

**(*R*)-Nicotine**
CAS 25162-00-9

Pharmacology not well characterized
7–20 $\times$ less potent than (*S*)-nicotine at nAChRs
Binds and inhibits AChE
50:50 of (*R*):(*S*) enantiomers in racemic nicotine

Fig 1. **Nicotine enantiomers and their pharmacological properties.**

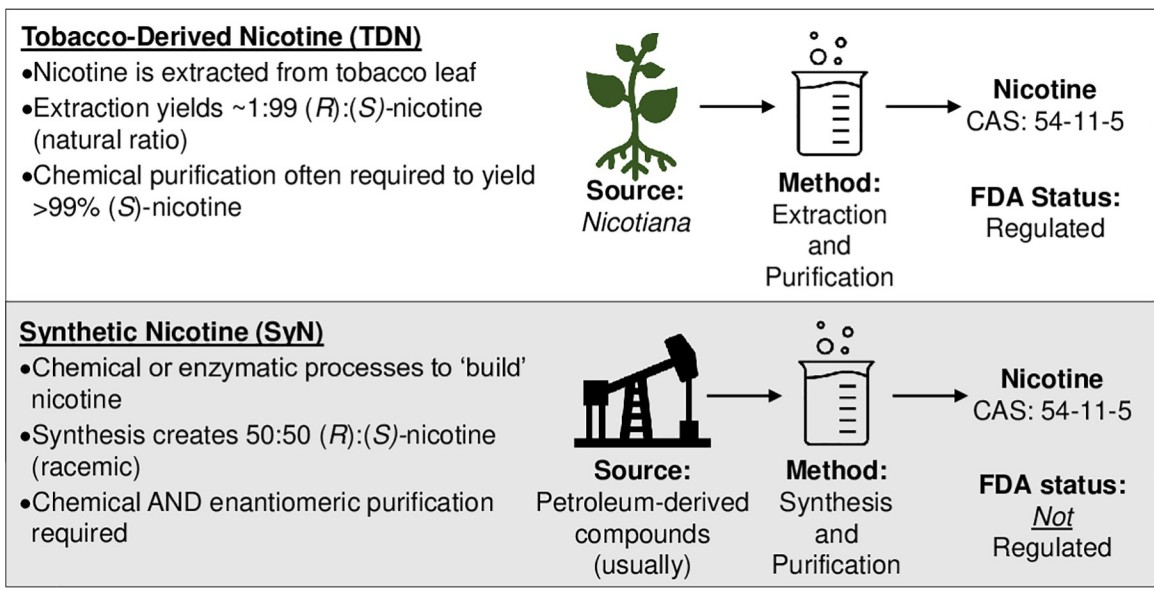

**Fig 2. Nicotine production pathways.**

standard analytical techniques. This makes nicotine source identification in a given product challenging. Analytical techniques that can resolve the two nicotine enantiomers, such as liquid chromatography, gas chromatography, NMR, and optical rotation, provide potential paths. These approaches are limited in that they can only distinguish SyN from TDN if the former is a racemic mixture. The two sources become indistinguishable If the nicotine used is $\geq$ 99% (S)-nicotine.

Beyond enantiomeric differentiation, TDN may retain tobacco signatures or impurities that would elucidate its tobacco leaf origins, such as nicotine degradants, tobacco-specific nitrosamines (TSNAs), metals, and other tobacco-related metabolites. Conversely, there could be synthetic process impurities that would indicate a synthetic origin, such as precursors or residual solvents. However, the level of purification that is now utilized for both types make this challenging and was recently commented upon by FDA CTP Director, Mitch Zeller, who said, "Tobacco-derived nicotine is now readily available as higher quality U.S. pharmaceutical-grade 99% nicotine, which no longer contains traditional tobacco agricultural markers like tobacco DNA or tobacco-specific nitrosamines–making it harder to distinguish tobacco-derived nicotine from synthetic" [11].

Given the high purity of both tobacco-derived and synthetic (S)-nicotine, it was proposed by Jordt that carbon isotope content analysis may provide a solution. Naturally abundant carbon exists as three isotopes, carbon-12 ($^{12}$C), carbon-13 ($^{13}$C), and carbon-14 ($^{14}$C or radiocarbon), with the latter being an isotope that undergoes radioactive decay. It is this property of $^{14}$C that is useful for the differentiation of biologically derived materials from fossil-derived (synthetic analogs). While $^{14}$C decays with a half-life of 5700 years, it is constantly replenished in the atmosphere (as $CO_2$) and incorporated into living plant matter at levels that are near constant. $^{14}$C in fossil-derived feedstocks (oil, gas), however, is not replenished and so, given the millions of years required for their formation, typically exhibit zero $^{14}$C levels. As such, the $^{14}$C content of a material can be used to determine if it is biologically derived, synthesized from petrochemical stocks, or even a mixture of the two. This technique, developed as ASTM-D6866 [19], is routinely used in the food and biofuel industry to confirm the authenticity of the stated origin and is determined by accelerator mass spectrometry (AMS) [20, 21].

The analysis of $^{13}C$ as the $^{13}C/^{12}C$ ratio, also known as $\delta C13$, is also possible via isotope ratio mass spectrometry (IRMS), typically in tandem with some form of chromatographic separation. However, this technique is generally more useful when there are clear metabolic differences between the origin sources, since it is based on the varying degrees of uptake of $^{12}CO_2$ and $^{13}CO_2$ by different plants, and it is frequently used to determine if various food products have been adulterated with other natural ingredients. Whether it would be applicable to distinguishing TDN from SyN is unclear, as no study has yet been performed to explore this.

The authors did not identify any peer-reviewed, published studies concerning the use of radiocarbon analysis to identify the source of a nicotine sample. In 2017, Next Generation Labs posted an authenticity study to their website that showed a 10% nicotine formulation claiming to be synthetically derived was actually tobacco-derived, since the $^{14}C$ content matched that of known TDN samples and not that of their own SyN product, TFN® Nicotine [22]. This non-peer-reviewed study indicates that it is feasible to use radiocarbon analysis to identify if a nicotine sample is tobacco-derived or synthetically-derived provided that it is isolated from the e-liquid formulation. Interestingly, they also showed that the test sample contained anatabine, which is formed in the tobacco plant, providing further evidence of the tobacco plant origin.

Herein, we present the results of our investigations into developing a robust method for distinguishing between TDN and SyN in nicotine-containing products. We explored a number of techniques to accomplish this, including the screening of nicotine samples for impurities such as nicotine degradants and metals, chiral analysis of nicotine and nornicotine, and radiocarbon analysis. The results from each technique will be presented and their significance and feasibility with respect to routine analysis discussed.

## Materials and methods

Nicotine (single production lots) and e-liquids samples were received and used as-is from suppliers and e-liquid companies (Table 1). ISO 17034 analytical standards were purchased from Spex Certiprep (Metuchen NJ, United States), Sigma-Aldrich (St. Louis MO, United States), Toronto Research Chemicals (North York ON, Canada), and Inorganic Ventures (Christiansburg VA United States). Internal standards were obtained from CDN Isotopes (Pointe Claire QC, Canada), Santa Cruz Biotechnology (Paso Robles CA, United States), and Inorganic Ventures. Reagents were sourced through Thomas Scientific (Swedesboro NJ, United States).

**Table 1. Nicotine samples and e-liquid formulations studied.**

| Nicotine Source | Identifier | Supplier | Description |
|---|---|---|---|
| Tobacco-Derived | TDN-1 | Alchem | (S)-Nicotine |
| | TDN-2 | North American Nicotine | (S)-Nicotine |
| | TDN-3 | Siegfried | (S)-Nicotine |
| | TDN-4 | TCI America | (S)-Nicotine, aged sample* |
| | TDN-5 | Puff Bar | (S)-Nicotine, 5% nicotine salt, Mixed Berries |
| Synthetic | SyN-1 | Next Generation Labs | TFN® (R)/(S)-Nicotine |
| | SyN-2 | Contraf-Nicotex Tobacco (CNT) | (S)-Nicotine |
| | SyN-3 | eLiquiTech | (S)-Nicotine, chemoenzymatic synthesis |
| | SyN-4 | Siegfried | (S)-Nicotine |
| | SyN-5 | Hangsen | (S)-Nicotine, aged 200 mg/mL formulation** |

* This nicotine sample had been stored at -20˚C but was extensively used over a two-year period.

** This formulation had been stored under ambient conditions for at least three months before testing began, and stored at -20˚C thereafter.

All testing was conducted at Enthalpy Analytical, LLC (Richmond, VA) or Beta Analytic, Inc. (Miami, FL) using validated methods under ISO 17025 accreditation, where applicable.

## Nicotine degradants by LC-MS/MS

The seven nicotine degradants (myosmine, cotinine, nornicotine, anabasine, anatabine, nicotine *N*-oxide, and *β*-nicotyrine) were determined using LC-MS/MS. Samples were prepared in triplicate at 50 mg/mL in a mixture of methanol/water (70:30) containing deuterated internal standards (myosmine-$d_4$, cotinine-$d_4$, nornicotine-$d_4$, and anabasine-$d_4$). Analysis was performed on an Agilent 1260 Infinity II HPLC system equipped with an Agilent 6460 Triple Quad mass spectrometer using a Waters XBridge C18 (2.1 x 50 mm, 2.5 μm) analytical column; mobile phase A: 0.1 M ammonium acetate (pH 10); mobile phase B: Methanol.

## Metals by ICP-MS

The nicotine samples were analyzed for thirteen metal analytes (arsenic, beryllium, cadmium, chromium, cobalt, copper, iron, lead, nickel, selenium, silver, tin, and zinc) by ICP-MS. Samples were prepared in triplicate through the microwave-assisted digestion of nicotine (0.5 g) in 2% aqueous nitric acid containing internal standards ($^{209}$Bi, $^{7}$Li, $^{72}$Ge, $^{103}$Rh, and $^{125}$Te). Extracts were analyzed using an Agilent 7700x ICP-MS system in helium gas mode for all analytes except beryllium (no gas) and selenium (hydrogen).

Palladium screening of the nicotine samples was also performed by ICP-MS. Samples were prepared in singlicate through initial digestion of nicotine in 1% aqueous nitric acid at 95˚C, followed by digestion in conc. nitric acid at 95˚C, and finally the addition of 30% hydrogen peroxide. Extracts were analyzed using an Agilent 7700x ICP-MS system in helium gas mode using $^{103}$Rh as the internal standard.

## Non-targeted analysis by GC-MS

Non-targeted analysis was performed by preparing nicotine samples at 5 mg/mL in ethanol containing internal standard (6-methylcoumarin). Samples were analyzed using an Agilent 7890 Gas Chromatograph coupled to a Mass Selective Detector (MSD) operating in full scan mode (35 to 450 amu). Any software-identified peaks were compared to spectra contained within the 2017 NIST/EPA/NIH Mass Spectral Library (NIST 2017).

## Chiral chromatography

All chiral chromatography was performed using an AZYP Nicoshell SPP column ($100 \times 4.6$ mm, 2.7 μm) using an isocratic elution profile with 0.2 wt. % ammonium formate in methanol. Flow rates and detection methods are detailed below.

Chiral analysis of the nicotine enantiomers was performed by UPLC-UV. Samples were prepared in singlicate at approximately 0.1 mg/mL in methanol. Analysis was performed using a Waters Acquity UPLC system equipped with a photodiode array (PDA), with a flow rate of 0.75 mL/min and monitoring at 260 nm.

Chiral analysis of the nornicotine enantiomers was performed by LC-MS/MS. Samples were prepared in singlicate at 50 mg/mL in methanol. Analysis was performed on an Agilent 1260 Infinity II HPLC system equipped with an Agilent 6460 Triple Quad mass spectrometer, with a flow rate of 1 mL/min. The flow was diverted to waste until one minute before the expected nornicotine elution time so as not to introduce concentrated nicotine into the mass spectrometer.

### Radiocarbon analysis

Sample preparation was performed by Enthalpy Analytical, LLC, and the subsequent radiocarbon analysis (ASTM D-6866 version 21) was conducted by Beta Analytic, Inc. (www.betalabservices.com).

Neat nicotine samples were analyzed as-is, whereas lab-made or commercial e-liquid formulations were first extracted to isolate the nicotine. Nicotine-fortified propylene glycol (PG)-vegetable glycerin (VG) (unflavored) formulations were dissolved in 1 M sodium hydroxide and extracted twice with hexanes. The combined organic extracts were dried over $Na_2SO_4$ and then evaporated to dryness. Flavored e-liquids were dissolved in 1 M hydrochloric acid and washed twice with dichloromethane. Next, the pH was adjusted to >10 with 5.5 M sodium hydroxide and then the solution was extracted twice with dichloromethane. After drying the combined organic extracts over $Na_2SO_4$, the solution was evaporated to dryness. If the resulting liquid was unscented and colorless (or very pale yellow), it was analyzed with no further processing. If the liquid was scented or colored, then it was redissolved in 5 mL of hexanes and washed twice with basic water. The hexanes were then dried over $Na_2SO_4$ and evaporated to dryness to yield a colorless or pale-yellow liquid that was then submitted for radiocarbon analysis.

Blends of tobacco-derived and synthetic nicotine were prepared by mixing the appropriate amounts of each type on a percent weight basis and analyzed as-is.

## Results and discussion

### Nicotine impurity screening

Due to the different production pathways for tobacco-derived and synthetic nicotine, it might be expected that the impurity profile of each would offer a potential means to distinguish the two sources. To this end, we screened a number of nicotine samples for common impurities that can be found in tobacco products (nicotine degradants and metals) and conducted a more expansive screen via non-targeted analysis that can identify unknown constituents through comparison of mass spectral data to the NIST Mass Spectral Library.

The U.S. Pharmacopeia monograph for nicotine lists seven nicotine-related compounds that must be analyzed and found to be ≤ 0.3 wt. % individually and ≤ 0.8 wt. % collectively in order to be considered acceptable for use [23]. These seven compounds, also known as nicotine degradants, are anabasine, anatabine, cotinine, nicotine-*N*-oxide, *β*-nicotyrine, nornicotine, and myosmine. Of these seven, anabasine and anatabine would be expected to be found exclusively in tobacco-derived nicotine, since the synthetic pathway would exclude their formation. Myosmine and nornicotine are both common intermediates in the chemical synthesis of nicotine and as such could potentially be more abundant in synthetic nicotine. Nicotine-*N*-oxide, *β*-nicotyrine, and cotinine are oxidation products of nicotine, and their presence would not necessarily be indicative of either production route. The sourced nicotine samples were analyzed for the seven nicotine degradants using Enthalpy's in-house validated LC-MS/MS method that was modified to provide lower detection and quantitation limits. The results for the four nicotine degradants that could be potential identifiers (anabasine, anatabine, myosmine, and nornicotine) are shown in Fig 3. Tabulated results for all seven nicotine degradants can be found in S1 Table.

All samples were observed to meet the U.S. Pharmacopeia criteria for nicotine-related compounds, but no clear trends in the analyte levels on the basis of their production source were apparent from the analysis. Anabasine and anatabine were quantifiable in two of the tobacco-derived samples, TDN-1 and TDN-4, but were not detectable in the remaining samples with

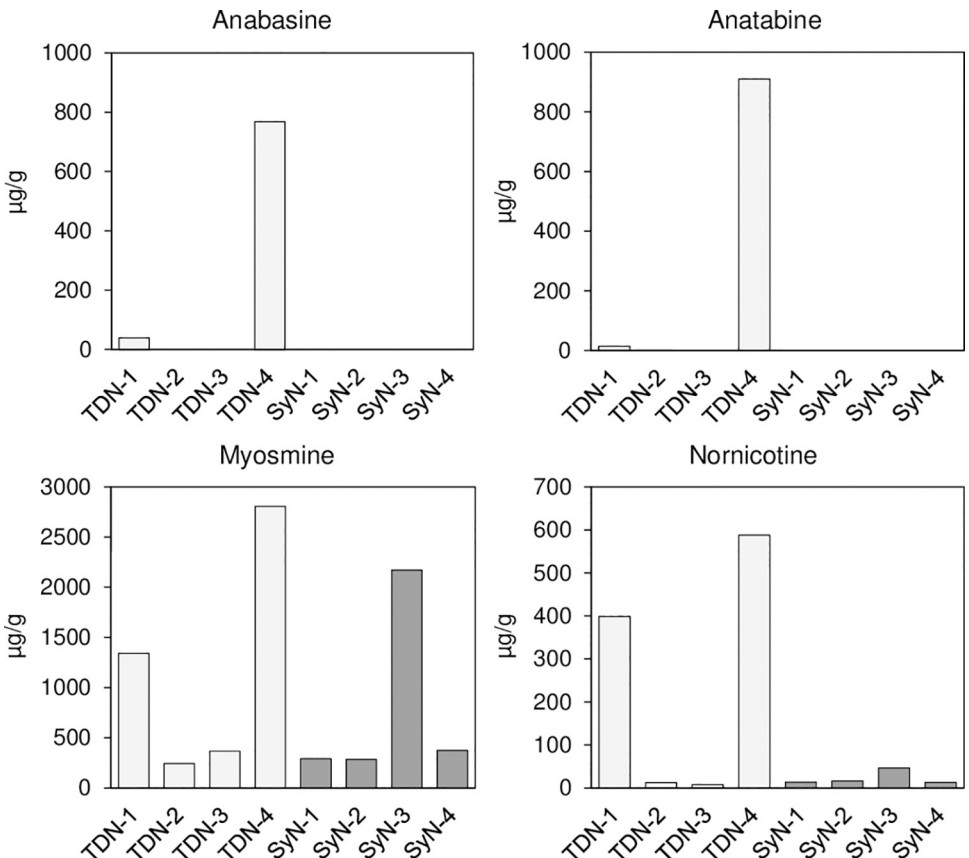

**Fig 3. Analysis of select nicotine degradants in tobacco-derived and synthetic nicotine samples.**

the exception of TDN-3 and SyN-4, for which anatabine was below the level of quantification (<0.5 µg/g). That TDN-2 and TDN-3 were comparable to the synthetic samples suggests these were purified to a greater degree than TDN-1 and TDN-4. Myosmine and nornicotine were observed in all of the studied samples to varying degrees, but again TDN-2 and TDN-3 were comparable to three of the synthetic nicotine samples, SyN-1, SyN-2, and SyN-4. SyN-3 exhibited higher levels of both myosmine and nornicotine, particularly the former, relative to the other three SyN samples. Cotinine and nicotine-*N*-oxide were observed in all samples, following a similar pattern of results to that seen for myosmine. *β*-Nicotyrine was seen only in TDN-4 and SyN-3. Based on the results, the only analyte that could possibly serve as an indicator would be anabasine. This analyte would be of limited utility, however, since anabasine was only detected in two of the four tobacco-derived nicotine samples. Furthermore, the analyzed samples were pure nicotine and even if anabasine were present in the nicotine itself, it would require extraction and/or concentration in order to detect anabasine in an e-liquid formulation.

Metals are known constituents of the tobacco leaf that can be retained in the final tobacco product [24], being introduced through either the soil or the air (via deposition onto the leaf). Geographical differences are also apparent based on the soil type, use of pesticides and fertilizers, and from environmental pollution [25, 26]. As such, there is the potential for metals to be present in tobacco-derived nicotine. Synthetic nicotine can involve the use of metal-based reagents during synthesis [14, 16], which could be present in trace amounts in the final product. To test these hypotheses, the nicotine samples were analyzed for various metals using

Enthalpy's in-house validated methods. The results, however, revealed no discernable trends, as the majority of metals assessed were not detected or were below the respective limits of quantitation (see S1 Table).

In order to expand the scope of our impurity screening, we conducted non-targeted analysis (NTA) of the nicotine samples to determine if there were any compounds that would be specific to a particular production route. The samples were prepared and analyzed using Enthalpy's in-house validated GC-MS method, with the resulting spectra compared against the 2017 NIST/EPA/NIH Mass Spectral Library. The only commonly observed impurity across all samples appeared to be cotinine, with no other compounds that were consistently observed between samples of the same production origin.

## Chiral chromatography

Chiral chromatography involves the resolution of enantiomeric mixtures through their differing interactions with a chiral stationary phase, thus allowing quantitation of the relative (or absolute) amounts of each enantiomer present. As mentioned in the introduction, the use of chiral chromatography to distinguish between TDN and SyN is not expected to be conclusive, since both types can be produced containing ≥ 99% (*S*)-nicotine. To confirm this, we analyzed (*S*)-nicotine and racemic (*R*)/(*S*)-nicotine standards of known purity by UPLC-UV using a modified version of a published method [27]. The AZYP Nicoshell SPP chiral column used for this purpose provides excellent baseline resolution between the (*S*)- and (*R*)-nicotine enantiomers (Fig 4(A)), allowing both to be accurately quantified. Next, we screened the neat nicotine materials (TDN-1 to 4 and SyN-1 to 4) and two formulations (TDN-5 and SyN-5) to determine their enantiomeric composition (Fig 4(B)). The analysis confirmed that all the test nicotine samples contained ≥ 99% (S)-nicotine, with the exception of SyN-1 which was found to be racemic as expected. On this basis, the chiral analysis of nicotine cannot definitively identify if TDN or SyN has been used in an e-liquid product, unless the amount of (*R*)-nicotine present greatly exceeds that found in TDN, i.e., > 1.5%.

The published method on which our chiral analysis was based [27] also demonstrated the separation of other nicotine-related compounds, such as anabasine, anatabine, and nornicotine. Given that all the nicotine samples we studied contained nornicotine to some degree, we explored the possibility of using this constituent to differentiate between TDN and SyN. Nornicotine in the tobacco plant is predominantly formed by enzymatic demethylation of nicotine [28, 29], a process that appears biased toward the (*R*)-nicotine enantiomer and leads to an observed wide variation in the (*R*)/(*S*)-ratio of tobacco-derived nornicotine (4–75%) [30, 31] which would not match the (*R*)/(*S*)-ratio of the nicotine from the same plant (0.1–1.2%) [32]. SyN is typically formed via methylation of nornicotine, which will result in the same (*R*)/(*S*)-ratio for both nicotine and nornicotine, even if further enantiomeric enrichment is performed. Therefore, it is hypothesized that chiral analysis of nornicotine may be able to distinguish between tobacco-derived and synthetic nicotine samples by characterizing and comparing the (*R*)/(*S*)-ratios of nornicotine and nicotine as they would be the same in SyN but different in TDN samples.

To test this theory, we analyzed the nicotine samples using LC-MS/MS and found that we could achieve excellent separation of (*R*)- and (*S*)-nornicotine (Fig 5(A)). However, the intensity of the observed signals was much lower than would be expected, with an up to 75% reduction in the total response (sum of both enantiomers) compared to the achiral method used for the nicotine degradant analysis. Furthermore, the quality of the chromatography varied considerably between the nicotine samples, with a number being too poor to allow confident analysis. Where the chromatography was acceptable, the measured results are shown in Fig 5(B)

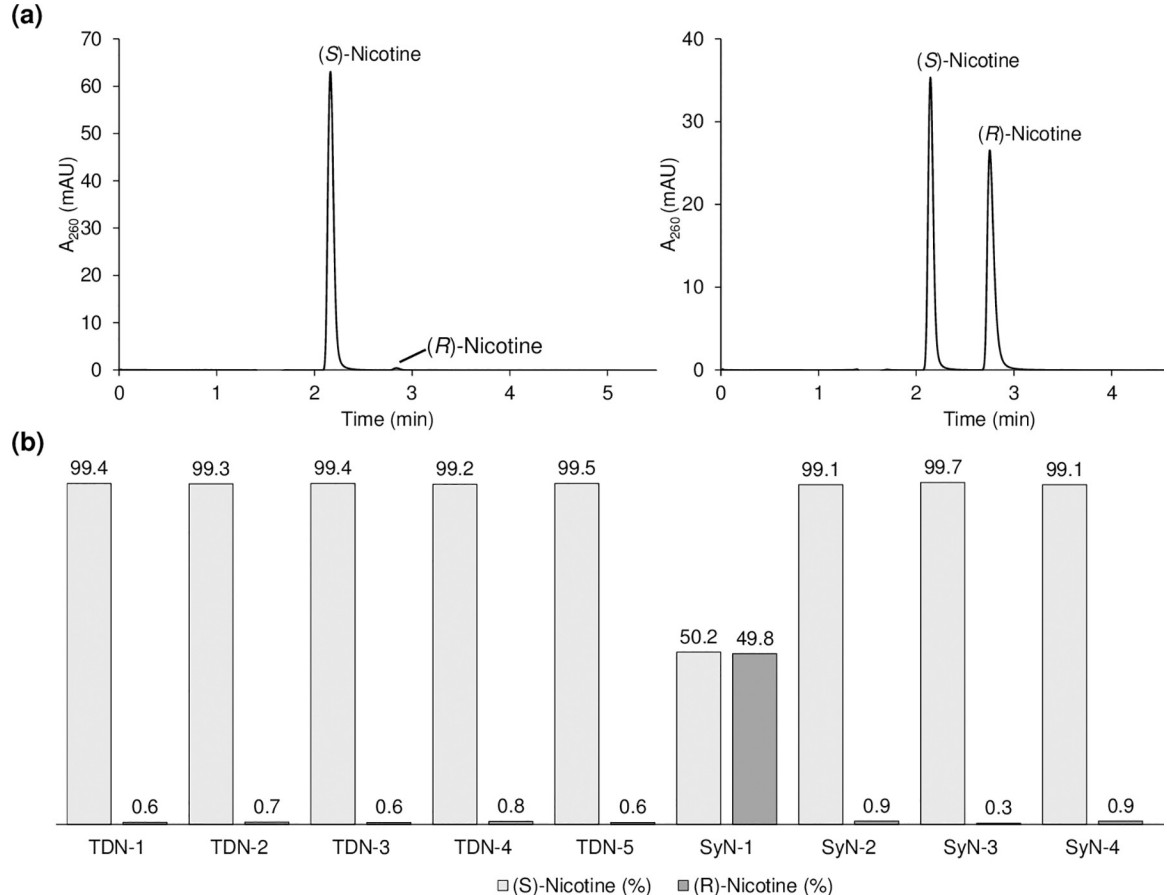

**Fig 4. Chiral analysis of the (*S*)- and (*R*)-nicotine enantiomers.** (a) Example chromatograms for an (*S*)-nicotine (left) and racemic (*R*)/(*S*)-nicotine (right) standards, and (b) relative amounts of (*S*)- and (*R*)-nicotine (as percentages) in various tobacco-derived and synthetic nicotine samples and formulations.

and would appear to support our hypothesis. Both TDN-1 and TDN-4 have nornicotine (*R*)/(*S*)-ratios that are mismatched to their respective nicotine (*R*)/(*S*)-ratios. SyN-1 and SyN-2, on the other hand, have well-matched nornicotine and nicotine (*R*)/(*S*)-ratios. The approach, therefore, appears promising but is currently hindered by poor sensitivity and matrix effects. Furthermore, given that the studied samples contained very small amounts of nornicotine, the highest being 0.06 wt. % in TDN-4, the analysis of e-liquid samples in which the nicotine and any nornicotine impurities are diluted in propylene glycol and glycerol would be even more challenging. Developing this into a routine analytical method, therefore, would require significant work and involve some form of extraction and concentration. The method applicability would also strongly depend upon the level of purification that the nicotine sample has been subjected, i.e., the amount of nornicotine present.

## Radiocarbon analysis

The analysis of a molecule's $^{14}$C content is perhaps the most definitive indicator of its origin as being biological, synthetic, or some combination thereof. The most commonly used method for assessing the radiocarbon content is the standardized method ASTM D6866, which uses accelerator mass spectrometry to separate $^{14}$C from the other two carbon isotopes ($^{12}$C and $^{13}$C). The result is typically given as "percent modern carbon" or pMC, in which the measured

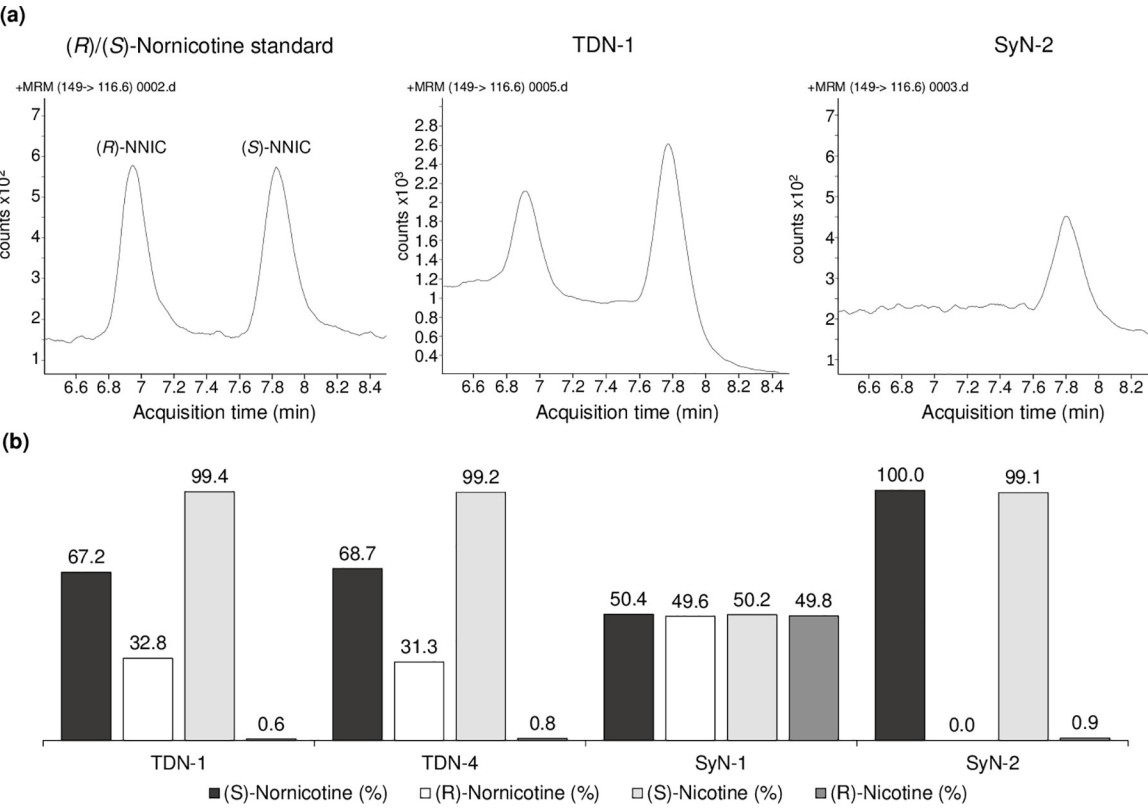

**Fig 5. Chiral analysis of the (*R*)- and (*S*)-nornicotine analysis in tobacco-derived and synthetic nicotine samples.** (a) Example chromatography in a racemic nornicotine standard, TDN-1, and SyN-2 (left-to-right), and (b) comparison of the (*R*)/(*S*)-nornicotine ratio to the corresponding (*R*)/(*S*)-nicotine ratio.

$^{14}$C content of the sample is normalized to the current atmospheric $^{14}$C levels. For greater clarity, the result is then often simplified to "% Bio-carbon," in which the results are presented on a 0% to 100% scale, since pMC can return results higher than 100 pMC. Substances that are purely biological in origin will give a result of 100% Bio-carbon, whereas purely petrochemical-based synthetic compounds will return a 0% Bio-carbon value. Synthetic materials that are derived from a mixture of biological and petrochemical feedstocks will fall somewhere in between depending on the relative amount of each source. Similarly, adulterated materials in which a natural substance has been mixed with a synthetic analog will fall between the two extremes. The radiocarbon analysis results from our nicotine samples are shown in Fig 6 and clearly indicate a distinct difference between the two production routes. The TDN samples are all 100% biobased, as expected, whereas the SyN samples returned values of 35% or 36% Bio-carbon. That the synthetic samples contained any $^{14}$C suggests that there is a common biologically-derived reagent being incorporated into the molecule at earlier stages of synthesis.

Given potential issues with the supply and demand of SyN, in addition to the higher associated costs, there is the concern of nicotine or nicotine products being sold that contain a mixture of TDN and SyN. For example, a company trying to skirt tobacco product regulations could add SyN to their TDN to raise the concentration of (*R*)-nicotine. Subsequent chiral analysis of products containing this nicotine mixture would appear to show the product contains SyN that has only been partially enantiomerically enriched. Theoretically, a mixture of the two nicotine types should give a radiocarbon result that is the sum of the proportion-weighted individual pMC values. We therefore tested this by preparing mixtures of known compositions

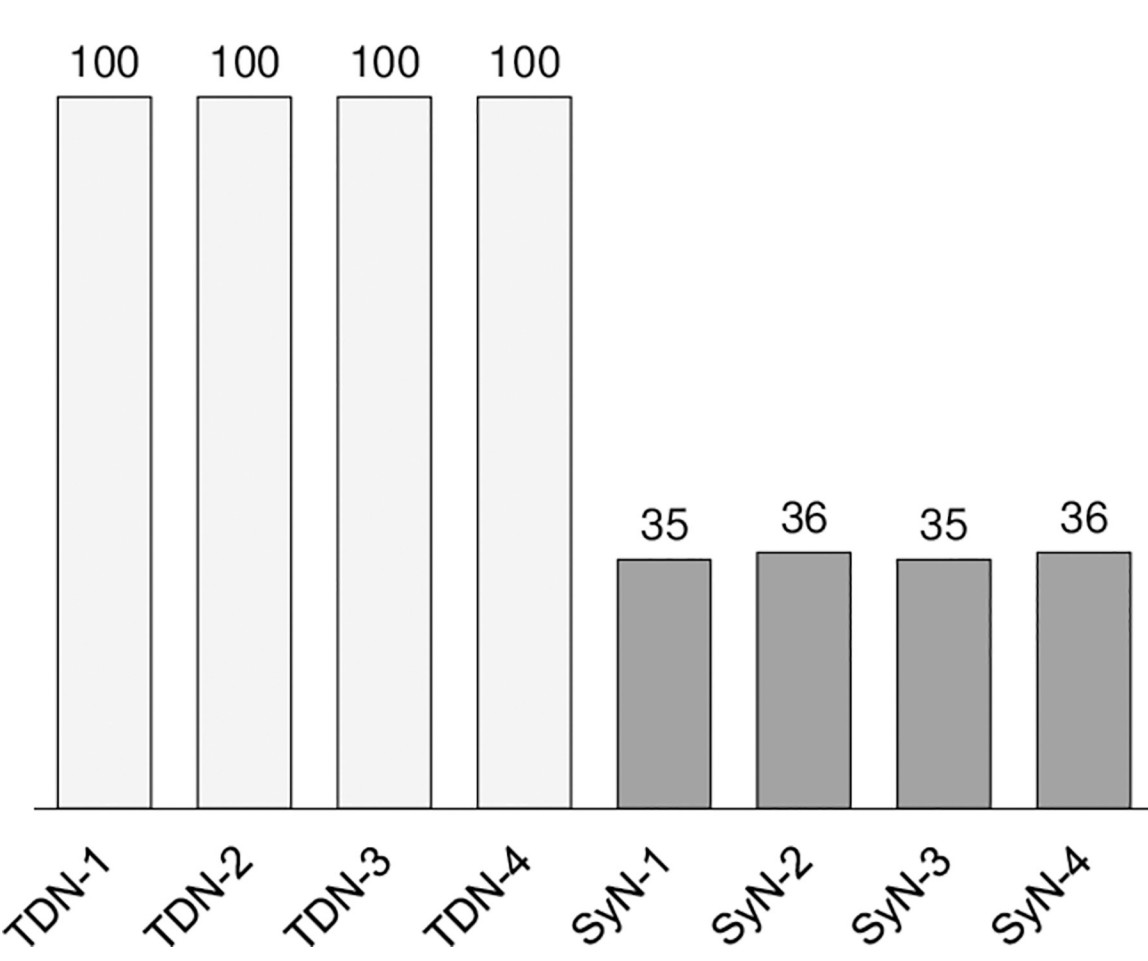

**Fig 6. Radiocarbon analysis of neat tobacco-derived and synthetic nicotine samples.**

using the TDN-2 and SyN-1. The radiocarbon analysis of these blends showed excellent agreement with the theoretical values (Fig 7) and confirms that the technique would be able to discern if a particular nicotine sample were purely tobacco-derived, synthetic, or a mixture of the two. However, it should be noted that this is predicated on the SyN used returning radiocarbon results that fall within the pMC value ranges we have observed during this study. The radiocarbon content of SyN could potentially be affected by the synthetic pathway and origin of the chemical ingredients used. Consequently, there may be SyN on the market, either currently or in the future, that does not possess a similar $^{14}$C content to those analyzed here. As such, it is recommended that the assessment of the relative amounts of TDN and SyN in a mixture of the two be for qualitative purposes only.

As determined in this study, the radiocarbon results from nicotine analysis can fall under one of three scenarios:

1. pMC < 40%: The test sample is confirmed to contain SyN.

2. pMC = 100%: The test sample is confirmed to contain TDN.

3. pMC value falls between those in scenarios 1 and 2: The result is suggestive of the sample containing a mixture of both SyN and TDN and warrants further investigation.

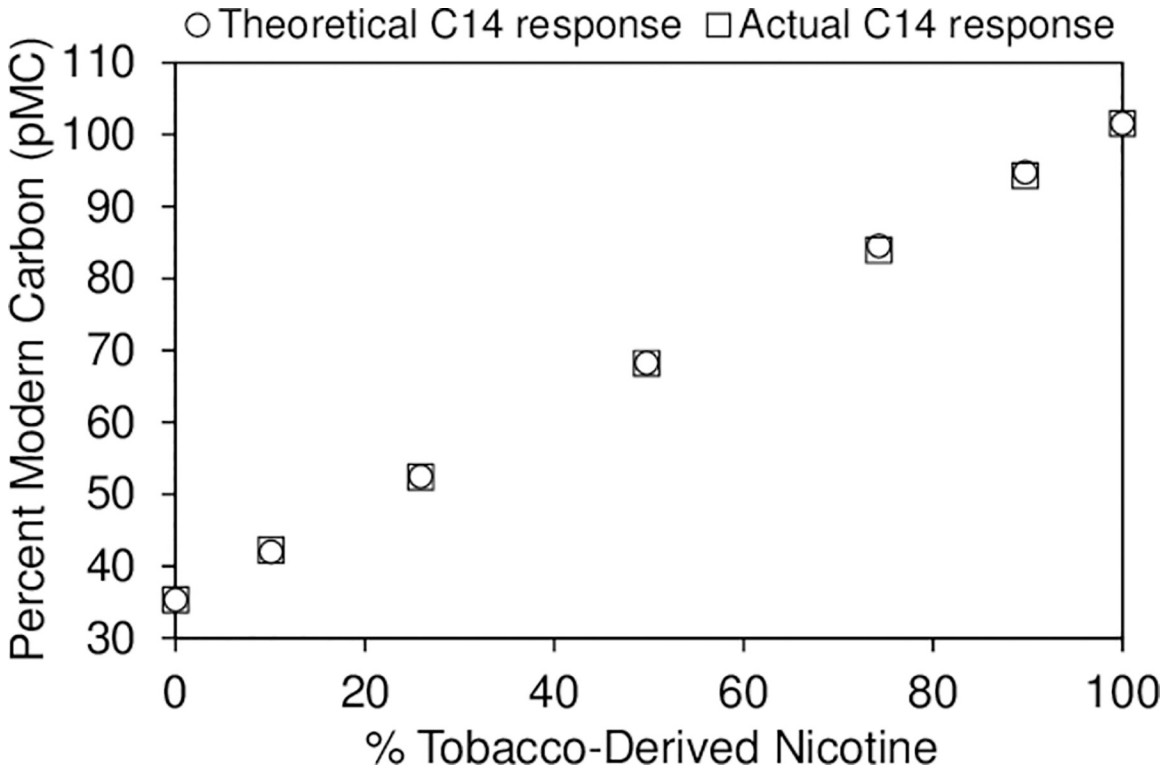

**Fig 7. Radiocarbon analysis of blended nicotine mixtures, comparing the observed pMC result to the theoretical value.**

From a regulatory standpoint, therefore, radiocarbon analysis of nicotine offers a definitive method for assessing the need for regulatory action with regards to nicotine products claiming to contain SyN. The caveat here, however, is that the nicotine must first be isolated from all other components of the product formulation, as discussed in the following section.

## Isolation of nicotine from ENDS e-liquid

While the radiocarbon analysis method is well-suited for determining the origin of pure materials, its main limitation is that it is not a selective separation technique. It is only capable of determining the $^{14}$C content of the sample that is analyzed and not a specific component of it. For this reason, the analysis of nicotine in an e-liquid formulation requires that the nicotine is first isolated from all the other components present, e.g., PG, glycerol (VG), flavorings, etc. If not, the result will be skewed depending on what ingredients are used. VG, for example, is quite often naturally derived and so if not removed would bias the result towards being 100% bio-based, while flavorings can be natural or artificial in origin. To be confident in the result obtained, therefore, an extraction method must be developed so that only the nicotine present is being tested.

The most obvious strategy to isolate the nicotine is to take advantage of its acid-base properties and use liquid-liquid extraction techniques. Indeed, the authenticity study posted by Next Generation Labs used this method to remove the nicotine from an unflavored PG formulation (10% nicotine). In our own studies, we found that the nicotine in similar unflavored formulations could be simply isolated by dissolving the e-liquid in basic water (pH > 10) and extracting with hexanes or dichloromethane, with no apparent extraction of either PG or VG into the organic phase. The addition of flavorings, however, introduces the need for extraction

**Table 2. Radiocarbon analysis of lab-prepared e-liquid formulation extracts.**

| Nicotine ID | Flavor | Flavoring Type | $^{14}$C Result in Extract |
|---|---|---|---|
| TDN-2 | None | N/A | 100%[a] |
| TDN-2 | Citrus | Natural | 100% |
| TDN-2 | Apple | Synthetic | 99% |
| SyN-1 | None | N/A | 36%[a] |
| SyN-1 | Citrus | Natural | 36% |
| SyN-1 | Apple | Synthetic | 35% |

[a] neat material result provided for ease of reference

under acidic conditions in order to remove these ingredients. We created our own e-liquid formulations using PG-VG (50:50) that had been flavored with natural or artificial flavors. These formulations were prepared using TDN-2 and SyN-1 at 3 mg/mL, giving four e-liquids that were first extracted as described above. The resulting nicotine extracts, however, were similarly colored to the e-liquids (pale green) and strongly aromatic, indicating that the flavorings were still present. The extracts were subsequently dissolved in 1 M hydrochloric acid and washed with dichloromethane, followed by pH adjustment and extraction into dichloromethane. The results of the subsequent radiocarbon analysis are shown in Table 2 and show excellent agreement with the nicotine used to prepare the e-liquid. The values for both of the e-liquids prepared using the artificial flavorings did appear to be slightly lower than expected for both nicotine types, suggesting that this flavoring was not completely removed during the extraction of these formulations. Nevertheless, the slight deviation is insufficient to cast doubt on the origin of the nicotine.

Using the modified extraction protocol, we next extracted the 200 mg/mL Hangsen synthetic nicotine e-liquid formulation, SyN-5. Some color changes were observed as the pH was adjusted, being pale pink in acidic conditions and yellow under basic conditions. The final nicotine extract was also colored, being a dark shade of pink. These observations are indicative of an ingredient not being fully removed and that it is quite possibly a food coloring given the apparent pH-sensitivity of the hue. To remove this component, the nicotine extract was dissolved in hexanes and washed with basic water to give a colorless liquid after solvent removal. Radiocarbon analysis of the sample showed the Hangsen nicotine to be 38% Bio-carbon, confirming its synthetic origins.

Our study has shown that it is quite feasible to extract the nicotine from e-liquid formulations and determine if it is tobacco-derived or synthetic. Adaptation of this technique to a more high-throughput environment, however, will require further optimization to both streamline the extraction workflow and to improve the recovery yield. The average yield of nicotine was approximately 40%, but was much lower for the dilute formulations (3 mg/mL) and consequently required a large sample volume to obtain a sufficient amount for radiocarbon analysis (> 50 mg nicotine). Extension of the method to oral non-tobacco nicotine products will also be important as the same current regulatory loophole could be similarly exploited to circumvent the costly PMTA process for these products.

## Conclusions

Radiocarbon analysis offers a definitive method to differentiate tobacco-derived nicotine from synthetic nicotine for regulatory purposes. This is especially true as most current TDN samples no longer have impurities or traditional tobacco agricultural markers like DNA or TSNAs, as they are removed using current extraction and purification techniques. Chiral analysis of

nicotine is an important complementary characterization to determine the enantiomeric purity of the sample which could have pharmacological and toxicological implications. If natural adulterants are suspected, other more traditional analytical methods are needed. These results should be pivotal in assisting regulators in determining whether products contain TDN or SyN and in assessing whether the products are misbranded in an attempt to skirt FDA tobacco product regulation. Future work will focus on refining analytical methods to improve method sensitivity as well as refining extraction techniques to extract and concentrate nicotine from low concentration matrix samples such as 3 mg/mL nicotine strength e-liquids more easily. Other tobacco products such as modern oral products (e.g., pouches, tablets, gum, discs) will also be analyzed in future work.

## Supporting information

**S1 Table. Tabulated analyte data for tobacco-derived and synthetic nicotine samples.** (PDF)

**S1 File. Raw data for nicotine degradants and metals analyses.** (XLSX)

**S1 Data.** (XLSX)

## Acknowledgments

The authors would like to thank the following for providing nicotine and e-liquid samples and helpful discussions: Anthony Dillion (Twelfth State Brands), Tricia Desmarais (My Vape Order, Inc.), Rob Reisel (Turning Point Brands), Kevin Burd (North America Nicotine), George Cassels-Smith (eLiquiTech), and Tony Nash (VapinDirect).

## Author Contributions

**Conceptualization:** Andrew G. Cheetham, Susan Plunkett, Bonnie G. Coffa, Stan Gilliland, III, Steve Eckard.

**Data curation:** Stan Gilliland, III.

**Formal analysis:** Andrew G. Cheetham, Susan Plunkett.

**Investigation:** Andrew G. Cheetham, Preston Campbell, Jacob Hilldrup, Bonnie G. Coffa, Stan Gilliland, III.

**Methodology:** Andrew G. Cheetham, Susan Plunkett, Preston Campbell, Jacob Hilldrup.

**Project administration:** Bonnie G. Coffa.

**Resources:** Stan Gilliland, III, Steve Eckard.

**Supervision:** Andrew G. Cheetham, Jacob Hilldrup, Bonnie G. Coffa, Steve Eckard.

**Validation:** Andrew G. Cheetham.

**Visualization:** Steve Eckard.

**Writing – original draft:** Andrew G. Cheetham, Susan Plunkett, Preston Campbell, Bonnie G. Coffa, Stan Gilliland, III.

**Writing – review & editing:** Andrew G. Cheetham, Susan Plunkett, Preston Campbell, Jacob Hilldrup, Bonnie G. Coffa, Stan Gilliland, III, Steve Eckard.

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
