## [Decision Letter · Decision Letter 0]

23 Feb 2022

PONE-D-22-00042Analysis and differentiation of tobacco-derived and synthetic nicotine products: Addressing an urgent regulatory issuePLOS ONE

Dear Dr. Cheetham,

Thank you for submitting your manuscript to PLOS ONE. After careful consideration, we feel that it has merit but does not fully meet PLOS ONE’s publication criteria as it currently stands. Therefore, we invite you to submit a revised version of the manuscript that addresses the points raised during the review process.

We look forward to receiving your revised manuscript.

Kind regards,

C. Michael Greenlief, Ph.D.

Academic Editor

PLOS ONE

Journal Requirements:

(No specific funding was provided for this work. The authors' salaries were provided by their respective employers, Enthalpy Analytical, LLC (AC, JH, BC, and SE) and Consilium Sciences (PC, and SG). SP is an independent consultant to both companies. All research costs were borne by Enthalpy Analytical, LLC. The funders played no role in the study design; collection, analysis, and interpretation of data; writing of the paper; and/or decision to submit for publication.

No part of this work has been funded by a tobacco company or similar entity.)

d) If you did not receive any funding for this study, please state: “The authors received no specific funding for this work.

3. The authors are paid employees of their respective companies. Enthalpy Analytical, LLC, a wholly owned subsidiary of Montrose Environmental Group, Inc., is an independent contract research laboratory with a focus on nicotine-containing products that provides analytical testing services for a wide range of clients, including tobacco manufacturers and government regulatory authorities. Consilium Sciences provides consulting services, offering scientific and regulatory solutions for materials science, nicotine, and cannabis organizations with a focus on potentially harm-reduced products.

5. Please amend the manuscript submission data (via Edit Submission) to include author Jacob Hilldrup. 

6. Thank you for stating the following in your Competing Interests section:  

(The authors are paid employees of their respective companies. Enthalpy Analytical, LLC, a wholly owned subsidiary of Montrose Environmental Group, Inc., is an independent contract research laboratory with a focus on nicotine-containing products that provides analytical testing services for a wide range of clients, including tobacco manufacturers and government regulatory authorities. Consilium Sciences provides consulting services, offering scientific and regulatory solutions for materials science, nicotine, and cannabis organizations with a focus on potentially harm-reduced products.) 

Additional Editor Comments:

Carefully consider the comments of Reviewer 1 and respond as needed.

Reviewers' comments:

Reviewer's Responses to Questions

**Comments to the Author**

1. Is the manuscript technically sound, and do the data support the conclusions?

Reviewer #1: Yes

Reviewer #2: Yes

2. Has the statistical analysis been performed appropriately and rigorously? 

Reviewer #1: Yes

Reviewer #2: Yes

3. Have the authors made all data underlying the findings in their manuscript fully available?

Reviewer #1: Yes

Reviewer #2: Yes

4. Is the manuscript presented in an intelligible fashion and written in standard English?

Reviewer #1: Yes

Reviewer #2: Yes

5. Review Comments to the Author

Reviewer #1: The manuscript, "Analysis and differentiation of tobacco-derived and synthetic nicotine products: Addressing an urgent regulatory issue," by Cheetham et al. is a timely and addresses an important issue (the identification of whether a sample containing nicotine is, in fact, synthetic or not. This is important both to the scientific and the regulatory audiences. The use of radiocarbon data is potentially useful and so this manuscript is a useful addition to the literature. However, at least in the view of this reviewer, there remains a constraint that the authors do not make clear to the reader: The residual number for %-bio-carbon (e.g., in Figure 5) for the SyN-# samples likely depends on the method of synthesis. Thus, the application of a calibration curve (e.g., Figure 6) may lead to an incorrect number for %-tobacco derived nicotine. Thus, their method is not as definitive as suggested, and this limitation should be made clear, both in the Discussion and the Abstract. Alas, the goal of a final and definitive method for characterizing all these samples remains elusive.

Reviewer #2: None

6. PLOS authors have the option to publish the peer review history of their article (what does this mean?). If published, this will include your full peer review and any attached files.

Reviewer #1: No

Reviewer #2: No

---

## [Author Response · Author response to Decision Letter 0]

14 Mar 2022

We are grateful to the Academic Editor and Reviewers who took the time to assess our manuscript and provide helpful feedback. Below are our point-by-point responses to the comments and suggestions that were made.

Responses to the Academic Editor Comments

Comment 1

Minor revisions were made to the manuscript formatting to be in compliance with the journal guidelines. One major revision was the replacement of Table 1 with a figure (Fig 1) due to the presence of graphical objects. The numbering of subsequent figures was adjusted accordingly.

Comment 2

The authors thank the Academic Editor for their timely response to our request for clarification on this comment. As suggested, the financial disclosure statement has been updated in the cover letter to be:

“No specific funding was provided for this work. No part of this work has been funded by a tobacco company or similar entity.”

Comment 3

No response required, see Comment 6.

Comment 4

All raw data for the analyses that were performed in triplicate (nicotine degradants and metals) have been included in the Supplementary Materials as an Excel workbook. Raw data for analyses performed in singlicate (chiral and radiocarbon analysis) were already included in S1 Table. The following statement has been added to the cover letter:

“All associated raw data can be found either in S1 Table or in the Excel workbook (Raw Data for Nicotine Degradants and Metals Analyses.xlsx).”

Comment 5

The manuscript submission data has been edited to include Jacob Hilldrup; we apologize for the inadvertent omission.

Comment 6

The authors thank the Academic Editor for their timely response to our request for clarification on this comment. As suggested, the Competing Interested statement has been updated in our cover letter to include the additional text indicated in red below:

“The authors are paid employees of their respective companies and do not claim any competing interests. Enthalpy Analytical, LLC, a wholly owned…”

Comment 7

The reference list has been reviewed and verified as requested. One change was the addition of a reference to a Politico article on synthetic nicotine that was released during the initial review period.

Responses to Reviewer 1 Comments

We thank the reviewer for their comments and appreciate that they feel the study will be of use to the scientific and regulatory community. Our responses to the specific points are given below:

Specific point 1

“However, at least in the view of this reviewer, there remains a constraint that the authors do not make clear to the reader: The residual number for %-bio-carbon (e.g., in Figure 5) for the SyN-# samples likely depends on the method of synthesis.”

The reviewer is correct that the %-biocarbon values for SyN samples may be dependent upon their route of synthesis and the reagents used, and as such may differ significantly from those we assessed. We have amended the text to clarify that different %-biocarbon values may be possible depending on the synthetic route and reagents used. Specifically:

Line 349:

“…of the two. However, it should be noted that this is predicated on the SyN used returning radiocarbon results that fall within the pMC value ranges we have observed during this study. The radiocarbon content of SyN could potentially be affected by the synthetic pathway and origin of the chemical ingredients used. Consequently, there may be SyN on the market, either currently or in the future, that does not possess a similar 14C content to those analyzed here. As such, it is recommended that the assessment of the relative amounts of TDN and SyN in a mixture of the two be for qualitative purposes only.”

We do note, however, that the samples we looked at represent the major suppliers of synthetic nicotine to the US market. Furthermore, since this work was performed, Enthalpy has tested additional samples for radiocarbon analysis and we have not seen any product with a %-biocarbon result that differs from those in this study, either being below 40% or at 100%. 

Specific point 2

“Thus, the application of a calibration curve (e.g., Figure 6) may lead to an incorrect number for %-tobacco derived nicotine.” 

The reviewer is correct in this assertion; however, it was not our intent that calibration curves would be generated and used to determine the SyN-to-TDN ratio of a mixture. Our intention for this experiment was merely to demonstrate that such a blended mixture would return a radiocarbon result that would be the proportion-weighted sum of their individual values. As such, a radiocarbon result somewhere between the two extremes we’ve observed would be suggestive of the test sample being a mixture of SyN and TDN, rather than one or the other. Taking into account the reviewer’s previous point, a SyN with a significantly different (and higher) biocarbon value could give a misleading identification as a mixture. Given that we have yet to observe such a SyN, however, the burden would be upon the manufacturer to prove they are using a genuine synthetic nicotine. The following text was added to clarify this:

Line 361:

“As such, it is recommended that the assessment of the relative amounts of TDN and SyN in a mixture of the two be for qualitative purposes only.”

 

Specific point 3

“Thus, their method is not as definitive as suggested, and this limitation should be made clear, both in the Discussion and the Abstract. Alas, the goal of a final and definitive method for characterizing all these samples remains elusive.”

If taken from the perspective of a method that could quantify the amount of TDN and SyN in a given sample, then the reviewer is correct. However, from the current regulatory standpoint it only needs to be shown that the sample in question does not contain any TDN to avoid regulatory action. The method as described allows for this since if the result is consistent with those of known SyN samples then it is confirmed to be synthetic in origin and no action can be taken. If any result above 40% is obtained, then it must be proven by the manufacturer they are using a genuine synthetic nicotine, otherwise a mixture would be assumed. A result of 100% would indicate the nicotine is tobacco-derived and the product has been misbranded (if labelled as SyN). From this regulatory perspective, we consider the method to be a definitive indicator with regard to the determining appropriate regulatory actions. The following text was added at the end of the “Radiocarbon Analysis” section to clarify this:

Line 358:

“As determined in this study, the radiocarbon results from nicotine analysis can fall under one of three scenarios:

1. pMC < 40 %: The test sample is confirmed to contain SyN.

2. pMC = 100 %: The test sample is confirmed to contain TDN.

3. pMC value falls between those in scenarios 1 and 2: The result is suggestive of the sample containing a mixture of both SyN and TDN and warrants further investigation.

From a regulatory standpoint, therefore, radiocarbon analysis of nicotine offers a definitive method for assessing the need for regulatory action with regards to nicotine products claiming to contain SyN.”

The first line of the “Conclusions” was also modified to read (changes highlighted in red):

Line 413:

“Radiocarbon analysis offers a definitive method to differentiate tobacco-derived nicotine from synthetic nicotine for regulatory purposes.”

---

## [Editor Report · Decision Letter 1]

1 Apr 2022

Analysis and differentiation of tobacco-derived and synthetic nicotine products: Addressing an urgent regulatory issue

PONE-D-22-00042R1

Dear Dr. Cheetham,

We’re pleased to inform you that your manuscript has been judged scientifically suitable for publication and will be formally accepted for publication once it meets all outstanding technical requirements.

Kind regards,

C. Michael Greenlief, Ph.D.

Academic Editor

PLOS ONE

Additional Editor Comments (optional):

The authors have addressed the minor concerns of the reviewers well. The manuscript is now in an acceptable form for publication.
---

## [Editor Report · Acceptance letter]

6 Apr 2022

PONE-D-22-00042R1 

Analysis and differentiation of tobacco-derived and synthetic nicotine products: Addressing an urgent regulatory issue 

Dear Dr. Cheetham:

I'm pleased to inform you that your manuscript has been deemed suitable for publication in PLOS ONE. Congratulations! Your manuscript is now with our production department. 

Kind regards, 

on behalf of

Dr. Charles Michael Greenlief 

Academic Editor

PLOS ONE